# Associations between Per- and Polyfluoroalkyl Substances (PFAS) and Cardiometabolic Biomarkers in Adults of Czechia: The Kardiovize Study

**DOI:** 10.3390/ijerph192113898

**Published:** 2022-10-26

**Authors:** Geraldo A. Maranhao Neto, Anna Bartoskova Polcrova, Anna Pospisilova, Ludek Blaha, Jana Klanova, Martin Bobak, Juan P. Gonzalez-Rivas

**Affiliations:** 1International Clinical Research Center (ICRC), St Anne’s University Hospital (FNUSA) Brno, 602 00 Brno, Czech Republic; 2Research Centre for Toxic Compounds in the Environment (RECETOX), Faculty of Science, Masaryk University, Kamenice 5, 625 00 Brno, Czech Republic; 3Foundation for Clinic, Public Health, and Epidemiology Research of Venezuela (FISPEVEN INC), Caracas 3001, Venezuela; 4Department of Global Health and Population, Harvard TH Chan School of Public Health, Harvard University, Boston, MA 02138, USA

**Keywords:** PFAS, cardiometabolic risk factors, population health, middle aged, elderly

## Abstract

Even though there is evidence of decreasing trends of per- and polyfluoroalkyl substances (PFAS) in Czechia, there are still major sources of PFAS pollution. Regarding the still-inconsistent results of the relationship between cardiometabolic health and PFAS, the present study sought to determine the association between PFAS levels and the presence of cardiometabolic biomarkers, including blood pressure and dysglycemia drivers in the Czech population. A cross-sectional study with 479 subjects (56.4% women, median: 53 years, range: 25–89) was conducted. Four PFAS were measured in serum: perfluorooctanoic acid (PFOA), perfluorononanoic acid (PFNA), perfluorodecanoic acid (PFDA), and perfluorooctane sulfonate (PFOS). The associations between natural log (ln)-transformed PFAS and cardiometabolic biomarkers were assessed through generalized additive models using linear regression and smoothing thin plate splines, adjusted for potential confounders. There were positive and significant (*p* < 0.05) associations between the ln-transformed PFOA and glucose (β = 0.01), systolic (β = 0.76) and diastolic blood pressure (β = 0.65); total cholesterol (β = 0.07) and LDL-c (β = 0.04); and PFOS with glucose (β = 0.03), BMI (β = 2.26), waist circumference (β = 7.89), systolic blood pressure (β = 1.18), total cholesterol (β = 0.13), and HDL-c (β = 0.04). When significant, the correlations of PFNA and PFDA were negative. Of the four PFAS, only PFOA and PFOS showed a positive association, even in serum levels not as high as the values from the literature.

## 1. Introduction

Since the 1940s, per- and polyfluoroalkyl substances (PFAS) have been produced and used in numerous commercial and industrial applications, including textile, carpet, and leather treatment (water and dirt proofing) and surfactants, firefighting foams, metal plating, and paper grease-proofing treatments [1]. PFAS can be released into the environment from wastewater treatment plants, landfills, recycling, and incineration plants and from the reuse of contaminated sewage sludge [2]. The general population is also exposed to background levels of PFAS through food consumption (mainly dairy products, fish, and meat), drinking water, and house dust [3].

PFAS are not easily degraded in the environment and are known as “Forever Chemicals” due to their extreme persistence [4]. PFAS will eventually transform into highly stable end products, remaining in the environment for hundreds or thousands of years [5], with environmental and health implications [6]. Most individuals living in industrialized countries exhibit one or more PFAS in their blood [2].

In Europe, there are approximately 100,000 PFAS-emitting sites [2], causing water contamination [7]. The PFAS regulation rules differ among European countries; for example, in Denmark, the use of PFAS in paper and board food packaging has been banned since July 2020, and no French fries bags exhibit PFAS; but in Czechia, these items recently showed highly persistent PFAS levels [4]. Despite the decreasing trends of PFAS in Czechia [8], sludge was recently detected PFAS contamination in 43 wastewater treatment plants (WWTP) [9], which is a major source of PFAS pollution [10].

Although there is no direct evidence linking PFAS exposure to mortality, there are several studies suggesting that PFAS exposure may be linked to deleterious health effects [11], including high total cholesterol [12], uric acid [13], glucose, insulin resistance [12,14], and blood pressure [15]. However, the association between PFAS and these biomarkers remains inconsistent, with different studies producing contradictory results [6,16].

Thus, considering this recent evidence and the inconsistent results regarding cardiometabolic health related to exposure to PFAS, the objective of the present study is to determine the association between PFAS levels and the presence of cardiometabolic biomarkers, including blood pressure and dysglycemia drivers in Czechia, using a population-based approach.

## 2. Materials and Methods

### 2.1. Study Design and Population

The Kardiovize study is a cross-sectional epidemiological study [17], including 2160 participants aged 25–64, and 270 subjects aged ≥65. The present study is based on a random subsample of 479 participants (279 subjects aged 25–64 and 200 participants aged 65 to 89 years. Study protocol complied with the Helsinki declaration and all participants signed the informed consent. The Kardiovize study was approved by the ethics committee of St Anne’s University Hospital, Brno, Czechia.

### 2.2. Data Collection

Face-to-face interviews were performed by trained nurses and physicians blind to study the hypothesis at the International Clinical Research Center of the St Anne’s University Hospital in Brno, using the web-based research electronic data capture (REDCap) [18]. The questionnaire included socio-demographics, (age, education, and marital status), cardiovascular risk behaviors (smoking status, nutrition, alcohol consumption, and physical activity), family and personal history, medications, hospitalizations, and mental health. The assessment also included blood pressure and measurement, anthropometry and venous blood sample.

### 2.3. Measurements

Blood pressure was measured with the patient alone using an automated office measurement device (BpTRU, model BPM 200; Bp TRU Medical Devices Ltd., Coquitlam, BC, Canada).

Laboratory analyses of cardiometabolic biomarkers were performed on 12 h fasting whole blood samples using a Modular SWA P800 analyzer (Roche, Basel, Switzerland); total cholesterol, triglycerides, and glucose were analyzed by the enzymatic colorimetric method (Roche Diagnostics GmbH, Mannheim, Germany); HDL-c was analyzed with the homogeneous method for direct measurement without precipitation (Sekisui Medical, Hachimantai, Japan).

The anthropometric assessment included height, weight, and waist circumference (WC). Height was measured using a stadiometer (SECA 799; SECA, GmbH and Co. KG, Hamburg, Germany). Body mass index (BMI) was calculated as weight (in kilograms) divided by height (in meters) squared. Educational level was categorized as primary, secondary, and higher. Household income was expressed in EUR/month and categorized as Low “<1200”, Middle “1200–1800”, or High “>1800”. Smoking status was classified as “non-smokers” or “current smokers” (smoking daily or less than daily during the past year). Participants were categorized into “non-drinkers” (including abstainers and those who had not drank in the previous 12 months) and “drinkers”. Alcohol consumption was assessed by the reported alcohol intake of the last week, expressed in the number of standard drinks. One standard drink was assessed as a glass of wine, bottle of beer, or shot of spirits, each corresponding to approximately 10 g of ethanol. Physical activity was assessed using the international questionnaire of physical activity (IPAQ) long version [19]. Subjects classified as “highly active” were those who participated in a vigorous-intensity activity at least 3 days per week, achieving a minimum of 1500 MET minutes/week, or 7 days per week in any combination of walking, moderate-intensity, or vigorous-intensity activities, achieving a minimum of 3000 MET minutes/week. Subjects classified as “moderately active” were those who participated in at least 20 min of vigorous physical activity 3 or more days per week, or at least 30 min of moderate-intensity physical activity or brisk walking 5 or more days per week, or 5 or more days per week of any combination of walking, moderate-intensity, or vigorous-intensity activities, achieving a minimum of 600 MET min/week. Subjects classified as “insufficient” were those who did not reach the activity levels listed above.

### 2.4. Laboratory Analyses of PFAS

The samples of 0.5 mL of frozen blood serum were processed and perfluoroalkyl substances were analyzed at a trace analytical laboratory of RECETOX accredited under ČSN EN ISO/IEC 17025:2018. Serum samples were heated to room temperature, homogenized using Vortex, and 200 μL of each serum sample was transferred to 96-well plate Phree Phospholipid Removal Plate (Phenomenex, Torrance, CA, USA). An amount of 5 ng internal standard (isotopic labelled compounds) and 600 μL of acetonitrile with the addition of 1% formic acid were used to each sample. Samples filtration was done and moved to glass vials, evaporated under nitrogen stream to last drop of solvent. An amount of 50 μL of methanol and 50 μL of NH4Ac in water were added. LC-MS/MS system—LC Agilent 1290 connected with QTrap 5500 (ABSciex, Framingham, MA, USA) with (ESI) equipped by chromatographic column SYNERGI 4μ Fusion MAX-RP 80 Ä 100 mm × 2 mm column (Phenomenex, Torrance, CA, USA), with pre-column Phenomenex SecurityGuard C18 4 × 2 mm. Mobile phase was following: 5 mM NH4Ac in water 55:45 (A) a methanol (B). Gradient elution was used.

Serum concentrations of four PFAS were measured: Perfluorooctanoic acid (PFOA), Perfluorononanoic acid (PFNA), Perfluorodecanoic acid (PFDA), and Perfluorooctane sulfonate (PFOS). For each compound, the limit of quantitation (LOQ) was set as 3 standard deviations (SD) of blank values, the limit of detection (LOD) as 1.5 × SD. PFOA (LOD = 0.020 ng/mL; LOQ = 0.070 ng/mL), PFNA (LOD = 0.004 ng/mL; LOQ = 0.012 ng/mL), PFDA (LOD = 0.004 ng/mL; LOQ = 0.010 ng/mL), and PFOS (LOD = 0.030 ng/mL; LOQ = 0.090 ng/mL) were detected in all samples.

### 2.5. Data Analysis

Continuous and categorical variables are expressed as a median and interquartile range, or proportion in percentage, respectively. Considering the data on PFAS markedly skewed to the right, they were naturally log (ln)-transformed in order to improve the normality of the data [20]. The Kolmogorov–Smirnov test was used to assess the normal distribution of the continuous cardiometabolic variables.

The generalized additive model (GAM) was used to evaluate potential non-linear relationships between PFAS biomarker levels and cardiometabolic variables adjusted by cofounders using thin plate regression splines. GAM is an adaptation that allows us to model non-linear data relationships while maintaining explainability. Covariates to be included as potential confounders were selected from available variables based on the literature. Pearson’s correlation coefficient and concurvity checked the multicollinearity among PFAS. Concurvity refers to the degree to which a smooth model term can be approximated by one or more smooth model terms. Like multicollinearity in a linear modeling framework, concurvity can complicate statistical inference using GAMs. All concurvity indexes are calculated on a scale of 0 to 1, with 0 indicating no concurvity and 1 indicating a total lack of identifiability [21].

The statistical analyses were performed using STATA software (version 14.0, StataCorp, College Station, TX, USA) and R (R Development Core Team 2010, R Foundation for Statistical Computing, Vienna, Austria. ISBN 3–900,051–07–0, URL: http://www.R-project.org/, accessed on 10 August 2022, specifically for GAMs through the R package mgcv [22]. The level of statistical significance was set at 0.05.

## 3. Results

### 3.1. Subject’s Characteristics

In total, 479 subjects were analyzed, with a median age of 53 years (30.4) and 56.4% being women. The median BMI was 26.0 (6.0), and the prevalence of cardiometabolic diseases were 15.1% T2D, 36.9% hypertension, and 47.8% high LDL-c (Table 1). Among the behavioral and social risk factors, 16.3% were smokers, 15.2% had insufficient physical activity, 87.7% reported alcohol consumption, only 3.3% had a low education level, and 55.5% had a low household income (Table 1).

Among the four PFAS evaluated, PFOS had the highest level (median 3.47 ng/mL), followed by PFOA (median 1.63 ng/mL), PFNA (median 0.58 ng/mL), and PFDA (median 0.18 ng/mL). The levels of PFAS by quartiles are presented in Table 2.

### 3.2. Analysis of Multicollinearity among PFAS

Table 3 presents the Pearson correlation coefficients of the relationship between the selected PFAS with each other. PFAS were significantly correlated with each other (*p* < 0.001). A correlation of 0.4 and higher represents concern about multicollinearity [23].

Table 4 presents the concurvity indexes. Assuming a conservative approach, as recommended by Johnson et al. [21], concurvity indices around 0.3 were accepted as the potential existence of multicollinearity. This threshold is arbitrary but successfully eliminates all model terms with Pearson’s r correlation around 0.50 or −0.50. Looking at the correlation and concurvity indices in Table 4 and Table 5, the majority of the concurvity was above 0.3, and most correlation coefficient values were above 0.4, supporting the concern of multicollinearity. Therefore, each PFAS was entered into its model.

### 3.3. Association of PFAS and Cardiometabolic Biomarkers

After adjusting by multiple covariates, blood glucose, BMI, waist circumference, blood pressure, total cholesterol, and HDL-c were significantly associated with PFAS blood concentrations (Table 5). Systolic and diastolic blood pressure, total cholesterol, LDL-c, and BMI were positively associated with PFOA (β = 0.76; *p* = 0.02, β = 0.65; *p* < 0.001, β = 0.07; *p* = 0.001, and β = 0.04; *p* = 0.04, respectively). There was also a positive association between PFOS and blood glucose, BMI, waist circumference, systolic blood pressure, total cholesterol, and HDL-C (β = 0.03; *p* < 0.001, β = 2.26 *p* < 0.001, β = 7.89; *p* = 0.001, β = 1.18; *p* = 0.01, β = 0.13; *p* = 0.01; β = 0.04; *p* = 0.002, respectively). On the contrary, BMI and waist circumference were negatively associated with PFNA (β = −1.22; *p* < 0.001, β = −4.37 *p* < 0.001) and PFDA (β = −2.92; *p* < 0.001, β = −10.12 *p* < 0.001), respectively. Diastolic blood pressure was only negatively associated with PFNA (β = −1.04; *p* < 0.001) and glucose and total cholesterol with PFDA (β = −0.06; *p* < 0.001, β = −0.09; *p* = 0.02).

## 4. Discussion

This analysis examined four PFAS and their association with cardiometabolic biomarkers in a population-based adult sample from Czechia. The results showed several associations between the evaluated PFAS and cardiometabolic biomarkers with a discrepancy in the direction of the relationship. The PFAS concentrations in blood serum were consistent with those reported in the examination of Czech national monitoring [24] and other epidemiological studies [25] but lower than those observed in occupational studies and highly exposed populations [13,16,25].

PFOA and PFOS are the most investigated PFAS, with strong evidence of their positive association with total cholesterol [16], similar to this study. On the contrary, the evidence of the associations between PFOA, PFOS, and other cardiometabolic biomarkers is mixed [12,25]. Our results showed a positive association of both PFOA and PFOS with blood glucose and systolic blood pressure, similar to the previous studies [25]. When compared to studies with similar age and median/mean blood level (ng/mL), the results from PFOS are analogous. For instance, Su et al. [26] in a study with 571 Taiwanese adults (20–60 years; PFOS: 3.2 ng/mL) presented a positive association between PFOS and glucose levels. Similar results were found by Liu et al. [27] in 1871 American adults (≥18 years; PFOS: 3.70 ng/mL). The positive association between PFOS and systolic blood pressure can be observed in 16,224 individuals from northern Italy [15] aged 20–39 years (PFOS: 4.63 ng/mL).

There was also a significant positive association between PFOS and both BMI and waist circumference. This relationship was previously described in a study also from China, including 1612 adults, that reported higher PFOS levels (24.2 ng/mL) than our study population [28]. The relationship between PFAS and HDL-c is also unclear. In a previous study from the US [29], PFOA was associated with HDL-c positively, as with the current results, while negatively based on another study from Italy [30].

Several negative associations of PFNA and PFDA with cardiometabolic biomarkers observed in our results have been previously reported in some other studies [31,32]. In 3629 American adults aged ≥20 years, there were inverse associations between triglycerides and PFNA and PFDA (respectively, 1.00 and 0.25) [31]. In a previous study from Italy, including 15,876 young adults aged 20–39 years, PFNA (0.53–0.58) was negatively associated with BMI [30]. Although the mechanisms by which PFAS may decrease the odds of certain biomarkers are unknown, the inverse association observed in this study may contemplate the anti-inflammatory and/or reduced insulin resistance effects of certain PFAS, as well as their potentially high oxygen-carrying capacity [33]. The PFAS structural similarity to fatty acids makes them increase the expression of genes involved in fatty acid oxidation in vitro [34], which could improve insulin sensitivity, stimulating free fatty acid storage and the use of glucose rather than fatty acids as an energy substrate [35,36]. Anti-inflammatory effects could be mediated by the activation of peroxisome proliferator-activated receptors (PPARs), which have anti-atherosclerotic properties, such as the suppression of vascular inflammation and oxidative stress [33].

The inconsistency of the literature suggests the need for a more comprehensive approach to assessing the association of PFAS with cardiometabolic health. To achieve higher accuracy of knowledge, one previous study suggested investigating the sex differences in the association instead of the inclusion of sex as the confounder [31], and another study focused on the potential mediating role of BMI [37], but none of these approaches changed our findings, and thus were not presented in the results of this paper.

The major limitation of the present study is the modest sample size; this, together with a large number of comparisons (four exposures vs. nine outcomes) makes the interpretation of the observed coefficients difficult. Studies of more representative samples are needed to corroborate the findings of this study in target subpopulations, which would give continuity to what would be the strength of the present research. The fact that associations can be detected not only in highly exposed cohorts (e.g., occupational) but also with values lower than those observed in most of the literature.

## 5. Conclusions

In conclusion, the PFAS exposure of the general population of Czechia is lower than in other more polluted areas of the world. The results indicated that even with lower PFAS levels, it was possible to detect associations with cardiometabolic biomarkers (several positive but a few negative), with results generally consistent with the literature. Future research should explore biological plausibility and mechanisms beyond the observed associations, which will enrich the existing lack of evidence on the effect of PFAS on cardiometabolic health in the general population.

## Figures and Tables

**Table 1 ijerph-19-13898-t001:** Characteristics of the subjects (*n* = 479).

Variables	Median (IQR)
Age	53 (30.4)
Men	43.6%
BMI (kg/m^2^)	26.0 (6.0)
Systolic Blood Pressure (mm Hg)	120.2 (24.4)
Diastolic Blood Pressure (mm Hg)	77.8 (11.8)
Glucose (mmol/L)	5.0 (0.9)
Triglycerides (mmol/L)	1.1 (0.7)
Total Cholesterol (mmol/L)	5.0 (1.4)
LDL-c (mmol/L)	3.0 (1.3)
HDL-c (mmol/L)	1.5 (0.5)
Type 2 Diabetes	15.1%
Hypertension	36.9%
Low HDL-c	10.4%
High LDL-c	47.8%
High Total Cholesterol	51.8%
High Triglycerides	16.1%
Educational Level	
Low	3.3%
Middle	57.0%
High	39.7%
Household income (Euro) (%)	
Low (<1200)	55.5%
Middle (1200–1800)	21.3%
High (>1800)	23.2%
Smoking (%)	16.3%
Alcohol (%)	87.7%
Physical Activity Level	
Insufficient	15.2%
Moderate	39.0%
High	45.7%
Medications	
Insulin	0.6%
Oral hypoglycemic	4.4%
Vasodilator	36.6%
Diuretic	15.0%
Hypolipidemic	18.6%

**Table 2 ijerph-19-13898-t002:** Descriptive statistics of perfluoroalkyl (PFAS) concentrations in blood of study participants (ng/mL).

	Minimum	25th	Median	75th	Maximum
Perfluorooctanoic acid (PFOA)	0.26	1.22	1.63	2.11	6.72
Perfluorooctane sulfonate (PFOS)	0.68	2.42	3.47	5.04	128.0
Perfluorononanoic acid (PFNA)	0.14	0.43	0.58	0.75	3.43
Perfluorodecanoic acid (PFDA)	0.01	0.14	0.18	0.28	1.34

**Table 3 ijerph-19-13898-t003:** Correlation between PFAS variables *.

	ln PFOA	ln PFNA	ln PFDA	ln PFOS
ln PFOA	1.00	0.69	0.40	0.50
ln PFNA	0.69	1.00	0.79	0.73
ln PFDA	0.40	0.79	1.00	0.69
ln PFOS	0.50	0.73	0.69	1.00

* All correlations were significant (*p* < 0.001).

**Table 4 ijerph-19-13898-t004:** Concurvity indexes between PFAS variables.

	ln PFOA	ln PFNA	ln PFDA	ln PFOS
ln PFOA	1.00	0.78	0.25	0.31
ln PFNA	0.78	1.00	0.65	0.58
ln PFDA	0.25	0.65	1.00	0.62
ln PFOS	0.31	0.58	0.62	1.00

**Table 5 ijerph-19-13898-t005:** Association between PFAS levels in blood and cardiometabolic biomarkers.

Cardiometabolic Variables	ln PFOA β (SE), *p*	ln PFNA β (SE), *p*	ln PFDA β (SE), *p*	ln PFOS β (SE), *p*
ln Glucose ^a^	0.01 (0.01), 0.32	−0.05 (0.03), 0.08	**−0.06 (0.02)**, **<0.001**	**0.03 (0.005)**, **<0.001**
BMI ^b^	−0.62 (0.88), 0.48	**−1.22 (0.15)**, **<0.001**	**−2.92 (0.30)**, **<0.001**	**2.26 (0.26)**, **<0.001**
Waist Circumference ^b^	0.13 (1.94), 0.95	**−4.37 (0.43)**, **<0.001**	**−10.12 (0.89)**, **<0.001**	**7.89 (0.79)**, **0.001**
Systolic Blood Pressure ^c^	**0.76 (0.32)**, **0.02**	−2.32 (2.19), 0.29	−1.65 (2.33), 0.48	1.18 (0.48), 0.01
Diastolic Blood Pressure ^c^	**0.65 (0.18)**, **<0.001**	**−1.04 (0.18)**, **<0.001**	−0.91 (1.64), 0.58	2.06 (1.06), 0.05
Triglycerides ^d^	0.01 (0.01), 0.31	−0.005 (0.01), 0.75	−0.01 (0.03), 0.69	0.01 (0.03), 0.74
Total Cholesterol ^d^	**0.07 (0.02)**, **0.001**	−0.02 (0.02), 0.46	**−0.09 (0.04)**, **0.02**	**0.13 (0.05)**, **0.01**
LDL-c ^d^	**0.04 (0.02)**, **0.04**	−0.01 (0.02), 0.72	−0.06 (0.03), 0.11	0.08 (0.05), 0.10
HDL-c ^d^	0.02 (0.01), 0.07	−0.04 (0.04), 0.32	−0.02 (0.02), 0.21	**0.04 (0.01)**, **0.002**

All bolded values are significant; ^a^—adjusted by age, sex, educational level, income, physical activity level, smoker, alcohol, body mass index, insulin, and hypoglycemic oral use; ^b^—adjusted by age, sex, educational level, income, physical activity level, smoker, and alcohol; ^c^—adjusted by age, sex, educational level, income, physical activity level, smoker, alcohol, body mass index, vasodilator, and diuretic use; ^d^—adjusted by age, sex, educational level, income, physical activity level, smoker, alcohol, body mass index, and hypolipidemic use.

## Data Availability

The data presented in this study are available upon request from the corresponding author. The data are not publicly available.

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
