# Peer review of "Associations between Per- and Polyfluoroalkyl Substances (PFAS) and Cardiometabolic Biomarkers in Adults of Czechia: The Kardiovize Study"

_ijerph, 2022, doi:10.3390/ijerph192113898_

Round 1

Reviewer 1 Report

Manuscript title: Associations between per- and polyfluoroalkyl substances (PFAS) and cardiometabolic biomarkers in adults of Czechia The Kardiovize study

Authours: Geraldo A Maranhao Neto1, Anna Bartoskova Polcrova1,2,, Anna Pospisilova1, Ludek Blaha2, Jana Klanova2, Martin 5 Bobak2 and Juan P. Gonzalez-Rivas 1,3,4

General comments

This starts out as a cross-sectional study of PFAS and cardiometabolic outcomes in a representative sample from the Czechia. Although at first glance the study is interesting, it is very shallow in its empirical review and its description of methods and so forth. Moreover, as it seems that all the references are wrong, I do not take this review seriously -  This seems to be a mistake somehow and I will not continue to review it until I am assured that the manuscript is accurate in its submission form.

Author Response

We apologize for the mistake. The references were from another study also submitted to the International Journal of Environmental Research and Public Health. Everything has already been fixed.

Reviewer 2 Report

The manuscript reports on an interesting study investigating the associations between PFAS and cardiometabolic biomarkers. PFAS see their wide applications in various industrial processes and daily products while the persistent nature of these chemicals also leads to growing public concerns, with reported environmental and health implications. While there are related studies reporting relevant investigations, it is acknowledged that the authors have made efforts in comparisons and discussion with relevance to these reports and extended their findings to strengthen the significance of the current study.

To enhance the manuscript, I have the following observations and comments suggested to the authors for further consideration:

1.

Ln 143: would seek the authors’ help for clarifying here or supplementing information in the manuscript on why perfluoroheptanesulfonic acid is being mentioned. I cannot get a clear understanding from the mentioning of “were detected in the all samples, and PFHpS”.

2.

More accurate descriptive statistics is required. For example, ln 184: according to Table 3 on p. 6, the p-value for HDL-c (PFOA) should be 0.002 instead of 0.02; ln 187: according to the statistics analysis on p.5, the negative association of diastolic blood pressure associated with PFNA should be beta=-1.04

3.

Would the validity details regarding the test/model be supplemented in ln 148-154, or may also consider to provide reference of the Kolmogorov-Smirnov test, or other mentioning as appropriate:

Massey Jr, F. J. (1951). The Kolmogorov-Smirnov test for goodness of fit. Journal of the American statistical Association, 46(253), 68-78.

4.

Suggestions for the referencing of the current manuscript:

Ln 115-116: for the International Questionnaire of Physical Activity (IPAQ), would consider the following or other suitable reference to be added:

Hagströmer, M., Oja, P., & Sjöström, M. (2006). The International Physical Activity Questionnaire (IPAQ): a study of concurrent and construct validity. Public Health Nutrition9(6), 755-762.

Ln 204-205: the descriptions regarding the indicated study by Su et al.[21] is not consistent with what is written in item 21 of the reference section (ln 319); similarly for the case of Liu et al. [22] (ln 206-207), which is inconsistent with item 22 in the reference section (ln 321).

References 34-51 on p.9, are they mentioned and explained in the main text?

5.

Abstract, ln 30-31: “Of four PFAS studied, two showed a positive association with most cardiometabolic variables…”, I suggest specifying here which two of the studied compounds are being described, for a clearer reference to readers in a preliminary reading of the abstract. Ln 3: in the article title: “:” is preferred than “.” after Czechia.

6.

Some other issues are also suggested:

1.

Further editing and checking is required. For example, there are spelling mistakes/wording issues, please refer to ln 19, “sough” should be “sought”; ln 73: it should be “The Kardiovize study was approved…” instead of “The Kardiovize was approved…”; ln 143: “in the all sample” should be “in all samples”; ln 166: “a” before 16.3% should be deleted;

Table 1 (p.5): the word “Insuficient” should be “Insufficient”;

Table 2 (p.5): the chemical should be “Perfluorooctance sulfonate” instead of “Pefluorooctane”;

 2.

I appreciate the authors for the mentioning of limitation and strength of the current study (ln 233-239). Would this part on limitation be further elaborated, or consider extending the major limitation in correlation for elaboration of future research suggestion. 

Author Response

Dear Reviewer, Thank you very much for your valuable suggestions. Answers are in the attached file.

Reviewer 3 Report

Reviewer's report ID ijerph-1888739

Title: Associations between per- and polyfluoroalkyl substances (PFAS) and cardiometabolic biomarkers in adults of Czechia. The Kardiovize study.

Date: September 29, 2022

The paper is devoted to evaluating the association between per- and polyfluoroalkyl substances (PFAS) levels and the presence of cardiometabolic biomarkers, including blood pressure and glucose in Czech population. Four PFAS were measured: Perfluorooctanoic acid (PFOA), Perfluorononanoic acid (PFNA), Perfluorodecanoic acid 23 (PFDA) and Perfluorooctane sulfonate (PFOS). The associations between natural log (ln) transformed PFAS variables and cardiometabolic biomarkers were assessed through generalized additive models using linear regression and smoothing thin plate splines, adjusted for potential confounders In the manuscript was found both positive and negative associations between PFAS components and cardiometabolic variables. I have some comments about the obtained results.

The comments about the manuscript:

1. The correlations between the logarithmic PFAC variables should be presented. Could it be that the negative correlation between ln PFNA/ln PFDA and cardiometabolic variables is due to the negative correlation between ln PFNA/ln PFDA and ln PFOA/ln PFOS? Hasn't there been an attempt to study the complex effects of several PFAS ingredients by including a few PFAS variables in the model?

2. If the including of a few PFAS variables in the model does not change the results, a negative association between some PFAS and cardiometabolic variables should be mentioned in Abstract and possible mechanisms should be discussed in more detail in the discussion section.

Conclusion: A major revision is necessary

Author Response

We appreciate the valuable contributions. Answers in the attached file.

Round 2

Reviewer 2 Report

I thank the authors for the substantial revisions made in the current manuscript.

I am satisfied with the authors’ efforts to address my comments in the previous review, and I would have a few observations for the team to further consider.

Ln 446-448 (p.5): the acceptance criteria for concurvity indices around 0.3 is mentioned. I would suggest the authors to write specifically at what exceeding level will the model terms will be rejected. The current description of using the phrase “… concurvity indices around 0.3 were accepted…” needs to be more specific for readers’ reference. For example, can take reference from the details in the reference 21 listed on p.9 (Johnston et al.), “We rejected model terms with concurvity indices that exceeded 0.3. This threshold was arbitrary but successfully eliminated all model terms in which Pearson’s r correlation exceeded 0.50 or -0.50, which we believe represents a conservative approach to model specification.”

Another comment is that looking at the correlation and concurvity indices in table 4 and 5 on p.5, majority of the concurvity is above 0.3 and for the correlation coefficients, there are also considerable portions of them having values above 0.4, the concerns of multicollinearity, which should be mentioned in the text. 

There are much efforts observed in addressing the previous comments related to references and editing. A further checking in these areas would be desirable. For example, ln 759, "...have not conflict of interest" should be "have no conflict of interest"; ln 763, why the name of the reference section is revised as "Uncategorized References"?

Author Response

The responses are attached.

Reviewer 3 Report

The authors corrected the manuscript according to the reviewer's comments

Author Response

Thank you for the valuable review.